# Photochemical Internalization: Light Paves Way for New Cancer Chemotherapies and Vaccines

**DOI:** 10.3390/cancers12010165

**Published:** 2020-01-09

**Authors:** Lara Šošić, Pål Kristian Selbo, Zuzanna K. Kotkowska, Thomas M. Kündig, Anders Høgset, Pål Johansen

**Affiliations:** 1Department of Dermatology, University Hospital Zurich & University of Zurich, Gloriastrasse 31, 8091 Zurich, Switzerland; lara.sosic@usz.ch (L.Š.); zuzanna.kotkowska@usz.ch (Z.K.K.); thomas.kuendig@usz.ch (T.M.K.); 2Department of Radiation Biology, Institute for Cancer Research, Norwegian Radium Hospital, Oslo University Hospital, Ullernchausséen 70, 0379 Oslo, Norway; selbo@rr-research.no; 3PCI Biotech AS, Ullernchausséen 64, 0379 Oslo, Norway; Anders.Hogset@pcibiotech.no

**Keywords:** photochemical internalization, photodynamic therapy, cytosolic delivery, cancer vaccination, cancer immunotherapy, cross-presentation, CTL

## Abstract

Photochemical internalization (PCI) is a further development of photodynamic therapy (PDT). In this report, we describe PCI as a potential tool for cellular internalization of chemotherapeutic agents or antigens and systematically review the ongoing research. Eighteen published papers described the pre-clinical and clinical developments of PCI-mediated delivery of chemotherapeutic agents or antigens. The studies were screened against pre-defined eligibility criteria. Pre-clinical studies suggest that PCI can be effectively used to deliver chemotherapeutic agents to the cytosol of tumor cells and, thereby, improve treatment efficacy. One Phase-I clinical trial has been conducted, and it demonstrated that PCI-mediated bleomycin treatment was safe and identified tolerable doses of the photosensitizer disulfonated tetraphenyl chlorin (TPCS_2a_). Likewise, PCI was pre-clinically shown to mediate major histocompatibility complex (MHC) class I antigen presentation and generation of tumor-specific cytotoxic CD8+ T-lymphocytes (CTL) and cancer remission. A first clinical Phase I trial with the photosensitizer TPCS_2a_ combined with human papilloma virus antigen (HPV) was recently completed and results are expected in 2020. Hence, photosensitizers and light can be used to mediate cytosolic delivery of endocytosed chemotherapeutics or antigens. While the therapeutic potential in cancer has been clearly demonstrated pre-clinically, further clinical trials are needed to reveal the true translational potential of PCI in humans.

## 1. Introduction

### 1.1. Cancer Therapy Development

Cancer therapy has evolved since Coley’s toxins and the birth of radiotherapy in the early twentieth century. The 1940s saw the development of chemotherapeutics, the first monoclonal antibodies for targeted therapies appeared in the 1980s, and photodynamic therapy (PDT), which is the topic of the current special issue of *Cancers*, was approved for treatment of cancer in the 1990s [1,2]. However, the first reports on photochemical treatment of cancer lesions date back to the early twentieth century [3]. Moreover, the complex relationship and crosstalk between tumors and the immune system has become gradually untangled and changed our understanding of oncology and cancer therapy. Therefore, the development of cancer immunotherapies has recently gained scientific and clinical momentum, with a giant leap made in 2011 as the FDA approved Ipilimumab ^®^for treatment of advanced melanoma. Ipilimumab is a monoclonal antibody targeting cytotoxic T-lymphocyte-associated protein 4 (CTLA-4) [4,5]. Second generation checkpoint inhibitors, such as programmed cell death protein 1 (PD-1) and programmed cell death 1 ligand 1 (PD-L1) inhibitors, have followed [6,7], and, today, more than 10 different immunotherapeutic agents, including checkpoint inhibitors, vaccine-based therapies, oncolytic viruses, and T-cell directed therapies for nearly 20 different indications across countless tumor types are available [1].

Despite these success stories, cancer therapy remains challenging, which is a major goal that is efficient and provides site-specific delivery of the therapeutic agents, the improvement of therapeutic outcomes, and the reduction of damage to healthy tissue. These problems are, in part, being addressed by defining and targeting cancer-specific molecules. In this case, personalized treatment can aid the design of a patient-tailored specific targeted therapy, which allows administration of the right treatment to the right person at the right time [8]. Therefore, the utilization of macromolecules is becoming increasingly relevant. However, since many of the therapeutic targets or receptors are intracellular, internalization to the cell cytosol is often crucial to achieve the expected biological effect [9].

### 1.2. Methods for Cytosolic Targeting

During the last three decades, several approaches have been suggested for the delivery of antigens and drugs to the cytosol. The mechanism by which a virus fuses with the cell membrane and hijacks the host protein production machinery has been one inspiration. Viral vectors and viral transcellular transduction proteins such as the twin-arginin translocation (TAT) sequence from the human immunodeficiency virus (HIV) or VP22 from the herpes simplex virus (HSV) have been widely used [10,11,12,13,14]. Since most viruses are immunogenic, tumor antigens have become more immunogenic when delivered with a viral vector [15,16]. In addition, recombinant viruses can be easily produced, administered, and quality-controlled [10]. As some tumor cells, e.g., ovarian cancer or lung adenocarcinoma, express folate receptors on their surface, folic acid was applied to facilitate delivery of chemotherapeutic agents [17] in patients with platinum-refractory epithelial ovarian, primary peritoneal, or endometrial cancer [18] and progressive lung adenocarcinoma [19]. Other pharmaceutical strategies for cytosolic targeting of drugs include cationic particles to shuffle antigens across the negatively charged cellular membrane [20,21], pH-sensitive and fusogenic liposomes that break up acidified phagolysosomes [22,23], and micelle-based immune-stimulating complexes (ISCOMs) that may facilitate antigen cross-presentation [22,24]. Very recently, reports on the use of photochemical internalization (PCI), which is a further development of PDT, has been suggested as a method to internalize cytotoxic therapeutics in tumor cells [25,26,27,28,29,30,31,32] or vaccine antigens in antigen-presenting cells (APC) [20,22,33,34,35,36,37].

### 1.3. Photodynamic Therapy (PDT)

Photodynamic therapy (PDT) is a well-established technique for the clinical treatment of several neoplasms such as non-melanoma skin cancer, esophageal cancer, non-small-cell lung cancer (NSCLC), bladder cancer, cervical cancer, head and neck cancer, breast cancer, pancreatic cancer, or prostate cancer [38], as described elsewhere in this issue of *Cancers*. Briefly, a photosensitizer is administered to the tumor lesion, and subsequent light activation induces photochemical energy, the generation of reactive oxygen species (ROS), which damage the cell membrane to cause cell death [39,40,41].

The physicochemical cytotoxicity mediated by PDT can further trigger inflammatory reactions and even tumor-specific adaptive immune reactions [42,43]. This immunological effect of PDT typically follows the production of damage-associated molecular patterns (DAMPs) by necrotic or apoptotic tumor cells, which are then recognized by APCs [44]. Activated APCs can present tumor antigens to T cells for stimulation of tumor-specific immune responses. Although the exact mechanism for such PDT-mediated immune effects is unclear, it has been demonstrated that PDT can elicit production of pro-inflammatory cytokines and anti-tumor immune responses [45,46].

### 1.4. PCI—A Photosensitizer—And Light-Driven Technology for Cellular Internalization of Molecules

Photochemical internalization (PCI) was developed as a method for light-enhanced cytosolic release of membrane-impermeable molecules that have been taken up by cells and entrapped in endocytic vesicles [47] (Figure 1). The molecules, e.g., chemicals, proteins, or nucleic acid DNA or RNA, together with the photosensitizer, get internalized by the cell via endocytosis or phagocytosis. This process leaves the internalized molecules entrapped within the lumen of the endosome [48,49]. PCI is based on amphiphilic photosensitizers that allow time-dependent dissociation from the outer plasma membrane, but not from the endosome. While the plasma membrane is light-insensitive after a certain time, the photosensitizer cannot escape the endosomal lumen or membrane, which, therefore, remains light sensitive. Light-induced ROS formation causes endosomal leakage with translocation of the internalized molecule into the cytosol for interacting with its designated target [37,50]. By these means, PCI has been demonstrated to enhance the therapeutic effects of a large number of molecules, including many types of macromolecules and some chemotherapeutic agents, that are subject to endosome-lysosome entrapment [48,50]. Hence, PCI is a method for intracellular delivery of molecules, but also a technology to enhance therapeutic specificity and the efficacy of drugs [9].

### 1.5. Photosensitizers in Use

There are several characteristics a photosensitizer drug should meet, e.g., specific tumor uptake, low toxicity in the absence of light, and long absorption wavelength. Longer wavelengths allow deeper tissue penetration [44]. Some of the most investigated photosensitizing drugs include hypericin [39], porfimer [38], and 5-aminolevulinic acid (ALA) [50]. The lifetime and diffusion distance of ROS are very limited, and the location of damage upon light activation is, therefore, highly dependent on the localization of the photosensitizer [25]. Porfimer sodium (Photofrin) was the first approved PDT agent (in Canada in 1993), and was FDA approved in 1995 for the palliative treatment of obstructive esophageal cancer [38]. The prodrug ALA or ALA esters (Metvix and Hexvix/Cysview) are the most widely used porphyrin-based photosensitizers for PDT. The topical application of these prodrugs leads to the production of protoporphyrin IX, which is very effective for photodynamic detection (PDD) and fluorescence guided resection of non-muscle invasive bladder cancer or for the treatment of thin and superficial skin lesions [50].

On the other hand, photosensitizers for PCI should be amphiphilic in order to enable non-receptor mediated endocytosis and dissociation of excess photosensitizers from the plasma membrane [29]. Examples of specific amphiphilic photosensitizers are disulfonated aluminum phthalocyanine (AlPcS_2a_) and disulfonated tetraphenyl porphyrin (TPPS_2a_) [48]. Disulfonated tetraphenyl chlorin (TPCS_2a_) is a second generation photosensitizer developed for clinical use in PCI [51]. In a recent clinical Phase-I trial, TPCS_2a_ was administered to patients with solid cutaneous or sub-cutaneous malignancies for internalization of the cytotoxic agent bleomycin [52]. Furthermore, a Phase-I dose-escalation study to assess the safety of fimaporfin-induced PCI of gemcitabine in patients with inoperable extrahepatic bile duct cancer (cholangiocarcinoma) and, based on the positive data, a pivotal Phase-II trial on extrahepatic biliary tract cancer has started recruiting patients [53]. Table 1 gives an overview of the photosensitizers currently in use.

### 1.6. PCI in Immunotherapy

Immunotherapies directed against cancer cells can broadly be divided into active, passive, or immunomodulatory. Passive immunotherapy involves the administration of tumor-specific lymphocytes or antibodies, whereas adjuvants or other immunologically active compounds can be immunomodulatory [4]. Checkpoint inhibitors operate by modulating the immune system’s endogenous mechanisms of T-cell regulation. They block co-inhibitory molecules on cytotoxic T lymphocytes (CTLs) and, consequently, debunk the inhibitory signals tumor cells elicit on T cells with promising results in clinical trials [54,55,56,57].

Cancer vaccines intend to elicit an active immune response by stimulating the body’s own immune system to target tumor-specific antigens. For recognition, such antigens can be presented by APCs to T cells by either intracellular or extracellular pathways. Extracellular pathogens and vaccines enter the APCs via the endosome or phagosome formation (Figure 2). The antigen uptake and the maturation of the antigen-containing endosome depend on the activation of pathogen recognition receptors such as Toll-like receptors (TLRs) and nucleotide-binding oligomerization domain-like or NOD-like receptors (NLRs). The endo- or phagosomes then fuse with lysosomes to form endo- or phagolysosomes in which the antigen gets loaded on major histocompatibility complex (MHC) class II molecules for presentation to CD4 T-helper cells. These cells typically provide help to B-cells for production of antibodies [20,58]. In that manner, the antigen never reaches the cytosol of the APC. Intracellular antigens, such as proteins from worn-out endogenous proteins, have direct access to the cytosol of the APCs. These antigens are processed by the proteasome to produce small peptide-fragments that can bind MHC class-I molecules in the endoplasmic reticulum. The peptide-MHC complex is transported to the cell surface for presentation to cognate T-cell receptors on CD8 T cells, which may then differentiate into CTLs. Since CTLs have been found to be one of the key players in fighting cancer [20,37,59], vaccination with tumor-associated antigens aims at inducing strong CTL immune responses. Hence, one major challenge is accessing the MHC class I pathway of antigen presentation. For this cross-presentation, antigens need to reach the cytosol of APCs, and this can be achieved by direct diffusion through the cell membrane or through endosomal escape after uptake.

Once it was demonstrated, PCI enabled internalization of chemotherapeutics into cells. The idea to deliver antigens to the cytosol of APCs for stimulation of MHC class I-restricted CD8 T-cell responses was born. Antigen and photosensitizer are co-administered, and subsequent endocytosis of antigen into photosensitized APCs enables ROS-mediated disruption of endosomes upon light treatment. Time is given for the amphiphilic photosensitizer to dissociate from the outer cellular membrane before light-activation. Lastly, the antigen is released from the endosomes into the cytosol where it can enter the MHC class I pathway of antigen presentation and stimulation of CTLs. Figure 2 illustrates PCI-mediated endosomal escape, proteasomal degradation, and CTL induction.

### 1.7. Cancer Vaccines

While prophylactic cancer vaccines can decrease the probability of late tumor development and have already been proven effective, the development of therapeutic vaccines against an established tumor is more difficult. Prophylactic vaccines against the human papilloma virus (HPV) and the hepatitis B virus (HBV) are clinically established to prevent cervical and oropharyngeal head and head and neck squamous cell carcinoma (HNSCC) or hepatocellular carcinoma (HCC), respectively [60,61]. The first and, so far, only approved therapeutic cancer vaccine is Sipuleucel-T, which is a dendritic cell (DC)-based cancer vaccine for the treatment of metastatic castration-resistant prostate cancer that increased overall survival by four to five months [4,62]. 

### 1.8. Objective

The objective of this paper is to systematically review research and development on the combined use of photosensitizers and cytotoxic therapeutics or antigens with the purpose of treating cancer.

## 2. Results

### 2.1. Study Selection and Study Characteristics

The literature search revealed 707 papers and the search on www.clinicaltrials.gov five trials. After removing duplicates, 636 publications were left, of which 433 were published from January 2009 through October 2019. After screening by title, 369 papers were excluded and 64 papers that met the eligibility criteria were screened by the abstract. Of those 64 records, 25 were read fully and 18 were included in the review. A PRISMA flow chart (Figure 3) illustrates the process of paper selection.

Table 2 summarizes nine preclinical papers on the use of PCI of cytotoxic therapeutics and two clinical studies where the PCI concept was tested. Table 3 summarizes seven preclinical studies on the use of PCI for cancer vaccination.

### 2.2. Preclinical Studies with PCI of Cytotoxic Therapeutics

For the use of PCI in the delivery of cytotoxic therapeutics, we reviewed three in vitro studies, five studies that combined in vitro and in vivo methods, and one in vivo study, i.e., nine studies in total. Olsen et al. showed in a doxorubicine-resistant cancer cell line that cross-resistance between PDT and doxorubicine could be overcome with PCI of a recombinant gelonin (rGel), a ribosome inactivating agent, and the photosensitizer TPCS_2a_ [25]. Since both doxorubicine-treatment and PDT-treatment led to the accumulation of ROS, the mechanism of PDT resistance in the doxorubicine-resistant MES-SA/Dx5 cell lines was suggested to be due to an increase in the production of ROS-scavenging enzymes. The measured effect was independent of caspase-3 activation, which proposed that the effects of PCI were apoptosis-independent. In PDT, the cytotoxicity is directly mediated through caspase-3-dependent ROS formation. Decreased p38 signaling was measured, as p38 is known to be an important death signal after TPCS2a PDT [66]. Hence, results indicate that in vitro toxicity in PCI of cytotoxic therapeutics was mediated by the delivered cytotoxic drug and not a PDT effect. Furthermore, PCI could circumvent ROS resistance.

O’Rourke et al. investigated the toxicity of PCI to dorsal root ganglion (DRG) neurons and associated satellite glia in vitro, as nerve damage is an impeding effect when treating malignancies within or close to nerves [26]. The neurons and glia were subjected to PCI treatment in a monolayer and in a 3-D culture system following incubation with bleomycin and the photosensitizers TPPS_2a_ or TPCS_2a_. The results showed that the head and neck squamous cell carcinoma cell line (PCI30) as well as satellite glia were more sensitive to PCI than DRG neurons and that the neurons survived PCI treatment under conditions sufficient to kill the tumor cells.

Martínez-Jothar et al. used two human breast cancer cells known as HER2-positive SkBr3 and HER2-negative MDA-MB-231, to show that PCI with the photosensitizer TPPS_2a_ and saporin-loaded polyester nanoparticles could specifically target HER2-positive cells when the particles were functionalized with the nanobody 11A4, which binds to the HER2-receptor [27]; the antineoplastic drug saporin is a ribosome inactivating protein. The results showed that PCI of cytotoxic therapeutics with the saporin-loaded nanoparticles selectively induced cell death of HER2-positive breast cancer cells.

Norum et al. tested PCI of bleomycin in a murine allograft model using T-cell efficient and deficient mice [28]. The mice received mouse colon carcinoma CT26. CL25 cells and the tumors were subsequently treated by the PCI of bleomycin. In wild-type immunocompetent mice, a curative effect was seen in >90% of the mice, whereas, in T-cell deficient athymic mice, only a tumor-growth delay with no curative effect was induced. Subsequently, when T cells from the spleen of PCI-bleomycin-treated thymic mice were isolated and adoptively transferred into naïve thymic mice, and the latter were challenged with CT26.CL25 cells, an inhibition of tumor growth was observed. Hence, photochemically internalized bleomycin was able to induce systemic and synergistic anti-tumor immunity to an extent that tumors inoculated immediately after treatment were rejected for up to two months.

Stratford et al. [29] as well as Bostad et al. [30] investigated the delivery of immunotoxins to CD133-positive sarcoma cells with stem-like properties. The immunotoxins were made by conjugating anti-CD133 antibodies with saporin. In human SW872 and HT1080 sarcoma cell lines, the PCI of CD133-saporin successfully targeted and killed CD133-expressing tumor cells. Surviving cells did not express CD133, had reduced proliferative capacity, and attenuated cancer stem cell properties [29]. When the surviving sarcoma cells were xenografted into immune deficient NOD-scid IL2Rγ0 mice, cells that were originally treated in vitro by PCI of CD133-saporin displayed greatly reduced tumor initiation and growth, compared to non-treated cells or cells treated with either PDT or PCI of saporin. Bostad et al. later demonstrated that PCI-induced targeting of CD133 could be achieved in vivo with a xenograft of human colorectal adenocarcinoma cells in mice [30]. In addition, targeting and killing effects were achieved in vitro using human cells derived from colorectal, breast, and melanoma cancers. Hence, both studies demonstrated selective and highly potent PCI of immunotoxins in stem-cell like tumor cells assuming sufficient surface expression of receptors for the immune ligand.

Eng et al. applied the same PCI concept to target the immunotoxin 225.28-saporin to highly aggressive breast cancer and melanoma cells that express chondroitin sulfate proteoglycan 4 (CSPG4) on the cell surface [32]. CSPG4 is known to promote chemotherapy and radiotherapy resistance and, furthermore, play an important role in tumor growth, but is hardly expressed in healthy tissue [67,68]. Three triple negative breast cancer (TNBC) lines (MDA-MB-231, MDA-MB-435, and SUM149) and two BRAFV600E mutated malignant melanoma cell lines (Melmet 1 and Melmet 5) were tested in vitro. PCI of 225.28-saporin induced a specific and strong synergistic cytotoxic effect in CSPG4-expressing cell lines. The effect was superior to both PCI of saporin and PDT only. The cytotoxic effect was dependent on light dose and CSPG4 expression. In athymic nude mice, PCI of recombinant fusion toxin scFvMEL-rGel effectively inhibited growth of A-375 melanoma. The cytotoxic response to PCI of 225.28-saporin in Melmet 1 cells was stronger than in the Melmet 5 cells, despite lower sensitivity to the chemotherapeutic agent dacarbazine or the BRAF inhibitor vemurafenib, and reduced expression of CSPG4 when compared to Melmet 5. As Melmet 1 and A-375 melanoma cell lines are mesenchymal-like, invasive, and de-differentiated, Melmet 5 is non-mesenchymal, non-invasive, and differentiated. The authors suggested that PCI may be effective in targeting cancer cells in a mesenchymal-like de-differentiated state.

Berstad et al. studied PCI for improving the delivery of drugs by targeting the epidermal growth factor receptor (EGFR) [31]. The researchers designed an EGFR-targeting fusion protein consisting of recombinant gelonin (rGel) toxin and EGF, and PCI of rGel-EGF was applied to EGFR expressing tumor cells in vitro and in vivo. PCI of rGel-EGF was highly effective against EGFR-expressing squamous cell carcinoma cells (SCC-026 and A-413) and even against cell lines resistant to the EGFR inhibitor cetuximab. The cytotoxicity was mediated by apoptosis, necrosis, and autophagy. In mice with A-431 xenografts, PCI of rGel-EGF caused reduced tumor growth, and, in mice with SCC-026 xenografts, reduced tumor perfusion and increased necrosis induction was observed after PCI of rGel-EGF as compared with control mice receiving PDT or rGel-EGF alone.

Vascular targeted therapeutics, e.g., bevacizumab, sunitinib, and sorafenib, are used as adjuvants to standard anti-cancer therapeutics by acting on vascular endothelial growth factor receptor (VEGFR)-signal transduction and inhibiting tumor angiogenesis [69,70]. Weyergang et al. showed that PCI of the fusion protein VEGF_121_/rGel allows targeting of specific VEGFR expressing tumor cells directly, as compared to indirect targeting by disruption of angiogenesis. The VEGFR1 expression on tumor cells has been associated with tumor growth and survival [71]. Two murine cancer cells were used known as the CT26.CL25 colon carcinoma and the 4T1 breast carcinoma cell line. PCI of VEGF_121_/rGel decreased tumor cell viability in both cell lines in vitro and led to tumor eradication in CT26.Cl25 allograft mouse models, as well as decreased tumor perfusion in both thymic and athymic mouse models. When mice were re-challenged with CT26.Cl25 cells three to six months after PCI treatment, no tumor growth was observed.

### 2.3. Clinical Studies with PCI of Cytotoxic Therapeutics

One clinical trial with PCI of cytotoxic therapeutics has been conducted so far [52]. In this Phase-I, dose-escalation, single-center study, the primary outcome was safety and tolerability of TPCS_2a_ in mediating PCI of bleomycin. A dose-limiting toxicity was recorded, according to Common Terminology Criteria for Adverse Events (CTCAE) scores. The maximum tolerated dose was defined as the dose at which 33% of patients within a cohort reported unacceptable toxicity. Secondary endpoints of the study were assessment of skin photosensitivity, pharmacokinetics of TPCS_2a_, and anti-tumor activity. The first-in-human trial recruited 22 patients of which 12 completed the three-month follow-up period. Of the 10 patients who did not reach the final visit, three did not reach day 28 (unrelated death), and seven left the trial after day 28 to receive other treatments. All patients were between 18 and 85 years old with local recurrent, advanced, or metastatic cutaneous or subcutaneous malignancies, which were clinically assessed as eligible for bleomycin chemotherapy. TPCS_2a_ was given intravenously and followed by 15,000 IU/m^2^ bleomycin intravenously on day four. Three hours after the bleomycin injection, the surface area of the tumor was illuminated (652 nm and 60 J/cm^2^). The initial TPCS_2a_ dose was 0.25 mg/kg. The dose was escalated in successive dose cohorts of three patients (0.5, 1.0, and 1.5 mg/kg).

No significant systemic adverse events (AEs) occurred after administration of TPCS2a. Upon light administration, Grade 1–2 AEs that occurred included localized erythema, localized swelling, nausea or vomiting, localized infection, photosensitivity skin reaction, pruritus, and localized sensory disturbance. Grade 3 AEs related to PCI were unexpected localized pain in cohorts 0.25 mg/kg (*n* = 3, 75%), 0.5 mg/kg (*n* = 4, 44%), and 1.5 mg/kg (*n* = 2, 67%). In the 1.5 mg/kg cohort, there was one Grade 3 localized infection (33%) and one Grade 3 photosensitivity skin reaction (33%). The latter was associated with edema and blisters on the back of the hands in a patient exposed to strong sunlight for prolonged periods against protocol recommendations. No treatment-related Grade 4 AEs and no treatment-related deaths were recorded. Mean pain scores were highest in the low-dose cohort (0.25 mg/kg) and immediately after light activation, as the patients were only given local anesthesia. In higher dose cohorts, pain was successfully managed by general anesthesia or sedation along with local anesthesia. Pain was recorded minutes after light exposure, escalated to a maximum short after, declined one to two hours later, and returned to clinically expected levels five to seven hours post light treatment. In all cohorts, pain was substantially reduced 24 h after illumination.

Independent of the TPCS2a dose administered, no adverse photosensitivity reactions were observed in patients subsequently exposed to 500 lux (approximately indoor light). For the 0.125 mg/kg cohort, photosensitivity was not even observed after exposure to 100,000 lux (approximately direct sunlight exposure). At TPCS2a doses of 0.25, 0.5, 1.0, and 1.5 mg/kg, increasing the number of mild (Grade 2) photosensitivity reactions were observed in patients exposed to 100,000 lux, and one moderate (Grade 3) reaction was observed in one patient that received the highest TPCS2a dose. Nearly the entire reaction disappeared within one day, but two out of six patients that received 1.0 or 1.5 mg/kg TPCS2a got palliative skin dressings for one week, and one patient from the 1.5 mg/kg cohort received additional antibiotics treatment.

The highest mean TPCS_2a_ concentration was recorded 30 min after administration. The mean values of AUC_0–∞_ increased with increasing doses. After a rapid first phase of elimination, TPCS_2a_ concentrations decreased toward baseline within 90 days, with one exception in the 1.5 mg/kg cohort, where the TPCS_2a_ concentration was higher on day seven than on day four. TPCS_2a_ was detected in the blood 90 days after administration in all cohorts. Since TPCS_2a_ was undetectable in urine in the first 14 patients, further urine sampling was stopped.

One case report describing the PCI of bleomycin has been published [65]. The 57-year-old Caucasian male was diagnosed with end-stage recurrent chondroblastic osteosarcoma of the mandible. The patient took part in the above mentioned Phase-I trial [52]. Prior to presentation, the patient had undergone chemotherapy, radiotherapy, and a number of surgical interventions. None of these were successful. The medical history included myocardial infarct with coronary stenting and treatment with aspirin, atorvastatin, amitriptyline, and morphine. Clinical examination revealed sarcoma affecting the right, middle, and lower face. The patient received 0.25 mg/kg TPCS_2a_, bleomycin, and light, as described above. Treatment was accompanied by a pain score of 9.9/10 for 2 h after illumination, dropping to 2.2/10 after 4 h. Three days post-illumination, histopathological analysis of the surgical biopsies showed extensive tumor necrosis with only scant viable tumor cells present. Further tissue shrinkage and necrosis was noted during the next three months, with biopsies confirming the tumor-free lesions. However, six months after therapy, the patient succumbed to cardiorespiratory failure after needing endoluminal carotid stenting and treatment of deeper tumor areas, mainly in the tongue base.

### 2.4. PCI Immunotherapy

Based on PCI of cytotoxic therapeutics, the idea emerged to apply the PCI to target antigens to APCs in order to stimulate anti-tumor immune responses. The first report on PCI-based vaccination was published in 2013, and since then, seven reports have been published, which are all reviewed in this paper. Five reports are combined in vitro and in vivo studies and two are solely in vivo studies. Five studies reported on the use of PCI in an allograft melanoma model. Two reports applied particle-based vaccines. One report described the delivery of small peptides, and one study reported the role of CD4 cells in PCI-based vaccination.

Waeckerle-Men et al. conducted a proof-of-concept study in 2013 where PCI of the antigen chicken ovalbumin (OVA) was performed in vitro using murine bone-marrow derived dendritic cells (DCs) and the photosensitizer TPCS2a [33]. After light application (435 nm and 13.5 mW/cm^2^), the DCs were co-cultured with murine splenocytes for analysis of functional antigen presentation to CD4 and CD8 T cells. PCI increased MHC class I-restricted antigen presentation as a function of the photosensitizer dose and light treatment. When the PCI-treated DCs were transferred to mice, this autologous immunization primed CD8 T-cell responses. The effect on antigen presentation was clearly dose-dependent. Cell death and apoptosis were also found to be TPCS_2a_-dose and light-dose dependent. Although in vitro testing showed that PCI increased MHC class I-restricted antigen presentation, PCI treatment induced only weak secretion of cytokines such as TNF-α, IL-6, IL-12, and IL-1β by APCs and it did not cause upregulation of the co-stimulatory molecules CD80 and CD86. 

In 2014, Håkerud et al. were the first to apply PCI-mediated immunization directly in vivo [34]. The TPCS2a-containing OVA vaccine was injected intradermally into the belly skin of mice. After different time points, different light doses were applied. The secondary study objectives were to determine appropriate photosensitizer and light doses as well as the timing of light administration. The authors concluded that an 18–24 h interval between vaccine and light administration was ideal in this mouse model. Shorter time intervals with the same TPCS2a doses caused erythema, wounds, and skin necrosis. An appropriate combination of doses was 50 µg TPCS2a and 4.86 J/cm^2^ light. Higher TPCS2a doses produced stronger immune responses, but also more AEs. Antigen delivery to cytosol of APCs was shown to be TPCS2a and light dependent via disruption of TPCS2a-containing and antigen-containing endosomes.

In 2015, the potential function of PCI-mediated vaccination in a murine model of melanoma was tested using OVA-expressing B16 melanoma cells as a model tumor and OVA as a model tumor antigen [35]. In a prophylactic vaccination model, mice first received the PCI-based OVA vaccines with light treatment 18 h later. On day five, the mice were challenged with B16-OVA-melanoma, and the subcutaneous grafting and growth of melanoma was monitored. PCI-based vaccination prevented tumor growth as compared to mice vaccinated without PCI. The proportion of mice surviving five weeks was higher in mice that received PCI-based vaccination (90%, *n* = 10) than in mice vaccinated without PCI (50%, *n* = 10) or left untreated (0%, *n* = 5). Similar data was observed when monitoring metastatic melanoma in the lungs. In a therapeutic model, PCI-based cancer vaccination reduced tumor growth and improved survival in melanoma-bearing mice as compared to the control mice. The protection was associated with increased proliferation and IFN-γ production by CD8 T cells, by tumor-infiltrating lymphocytes (TILs) of the CD8 subset, and by tumor-infiltrating macrophages (TIMs). The process of PCI was independent of MyD88 and TLR4 signaling, but dependent on trypsin-like and caspase-like proteasome activity and partly on chloroquine, at least in vitro. The chloroquine results suggests that PCI-mediated cytosolic targeting of antigen can also be a result of translocation from late endosomes or phagolysosomes.

Bruno et al. demonstrated that PCI-based vaccination could be performed with antigen and photosensitizer contained in a particulate antigen delivery system composed of poly(lactide-co-glycolide) (PLGA) microparticles of approximately 1 µm in size [22]. Mice immunized with particles containing OVA and TPCS2a showed stronger CD8 T-cell proliferation than mice immunized with PLGA particles containing OVA antigen without TPCS2a. Moreover, the production of pro-inflammatory IFN-γ, TNF-α, and IL-2 by CD8 T cells was improved upon PCI-based immunization compared with PCI-free immunization. CTL responses and granzyme B production were facilitated. Importantly, when mice were challenged with B16-OVA melanoma cells three weeks post immunization, no tumor growth (0%, *n* = 8) was observed in mice that received PCI-based vaccination, whereas a solid melanoma tumor was observed in 71% (*n* = 7) of mice immunized without PCI, and 75% (*n* = 8) of tumor grafting was observed in non-immunized mice. The study further showed that PCI was associated with reduced MHC class II-mediated antibody responses, which indicates a successful shift from MHC class II to MHC class I antigen presentation.

Hjálmsdóttir et al. also applied PCI with vaccine particles, using liposomes prepared from dipalmitoyl phosphatidylcholine and cholesterol [20]. Mice received either OVA liposomes or OVA liposomes mixed with TPCS2a liposomes, and antigen-specific T-cell and B-cell responses were analyzed ex vivo by flow cytometry and enzyme-linked immunosorbent assay (ELISA). PCI-based immunization improved CD8 T-cell responses and reduced B-cell antibody responses as compared to PCI-free immunization, which, again, suggests a shift from MHC class II to MHC class I antigen processing and presentation. Additionally, the study demonstrated that particles can protect the photosensitizer from light-induced inactivation during storage.

Haug et al. used PCI for vaccination with short peptides [36]. In vitro, the PCI of peptide antigens in APCs promoted cytosolic antigen delivery and resulted in increased formation of MHC class I-peptide complexes on the APC surface. The subsequent stimulation of antigen-specific CD8 T-cell proliferation was 30-fold to 100-fold more efficient when compared to vaccination with the peptide alone. Lastly, the authors showed successful antigen-specific CTL activation in mice after photochemical internalization of peptides from the human papilloma virus (HPV 16 E7, HPVaa43–78) and melanoma tyrosinase-related protein 2 (TRP aa180–188).

Since the contribution of CD4 T-helper cells are considered important in the priming of cytotoxic CD8 T-cell responses, and, since the mechanism of action of PCI-based vaccination involves the stimulation of MHC class I-restricted CD8 T-cell responses at the expense of MHC class II-restricted CD4 T-cell responses, Varypataki et al. most recently studied the role MHC class II and CD4 T helper cells during PCI-based vaccination in mice [37]. While MHC class II expression or CD4 T helper cells were indispensable for non-PCI vaccination, PCI-mediated vaccination did not require MHC class II expression or CD4 T helper cells for stimulation of CD8 T-cell responses. Moreover, the study also demonstrated that therapeutic vaccination with PCI of soluble antigens or of lethally irradiated melanoma tumor cells significantly prolonged progression-free survival of mice with established melanoma. The effect was independent of CD4 T helper cells.

## 3. Discussion

### 3.1. PCI of Cytotoxic Therapeutics

PDT kills tumorous tissue by means of photosensitizer and light. In contrast, photosensitizer and light in PCI is not primarily used to kill tumor cells but as a vehicle for specific and intracellular delivery of anti-cancer drugs. By existing data, PCI has proven to be a promising method for targeting therapeutic molecules to tumor cells for the purpose of specific killing. A wide range of drugs have been tested, e.g., macromolecular proteins, peptides, nucleic acids, and synthetic polymers, but also low-molecular weight chemotherapeutic drugs [47,49,50,72]. The method of PCI of cytotoxic therapeutics is especially applicable to drugs where the therapeutic target is intracellular and the PCI mediating cytosolic delivery of drugs has poor access to their cytosolic target.

One potential application of the PCI of cytotoxic therapeutics is to overcome drug resistance, which is one of the major challenges to reach effective cancer treatment. Up to 50% of malignant tumors are intrinsically resistant to chemotherapy [73], with the additional problem of attained resistance after repetitive drug administration. In this case, PDT represents an alternative treatment method that is usually not associated with resistance. However, PDT is often tissue and cell unspecific and, therefore, mostly applicable for superficial and solid tumors (Table 1). By contrast, the combination of PDT and chemotherapy in PCI has been suggested to enable PDT-guided delivery of chemotherapeutic drugs to specific tumor cells, and, thereby, overcome the problem of resistance [50]. For example, the chemotherapeutic drug bleomycin is approved for treatment of testicular carcinomas, lymphomas, head and neck cancers, and other non-melanoma skin cancers, but is known to become trapped in intracellular compartments after administration, which consequently leads to the need for higher therapeutic doses [74]. This is associated with an increased risk of pneumonitis and subsequent lung fibrosis [75]. However, PCI of bleomycin enhanced cytotoxicity compared to bleomycin alone both in vitro [26] and in vivo [28], which suggests improved bio-distribution and organ specificity with reduced resistance. By consequence, PCI can enable reduction of the dose needed to achieve a therapeutic effect and thereby reducing non-specific, adverse events [76].

The use of PDT for cell-targeted delivery of immunotoxins, such as the ribosome-inactivating proteins saporin and gelonin, have also been investigated with PCI. Immunotoxins were coupled with specific cell-targeting proteins that can bind to CSPG4 [32], to CD133-expressing cancer stem cells [29,30], to EGFR [27,31], or to VEGFR [63]. Targeting of CD133-expressing cancer stem cells is based on the knowledge that a small population of stem-like cancer cells are often resistant to traditional chemotherapies, where the consequence is tumor relapse and metastasis [77]. Unfortunately, the clinical success of CD133-directed immunotoxins has been compromised by the potential harm on normal stem cells that also express CD133 [78,79,80,81]. However, the PCI of CD133-directed immunotoxins seems to pose a potential solution to normal stem cell toxicity by increasing tumor-cell selectivity [29,30]. Systemically administered CD133-directed immunotoxins were found to localize predominantly in the tumor tissue, with no detection in normal tissue except in the kidney and the liver [30]. Hence, PCI may reduce the frequency of drug administrations, which by consequence may reduce treatment-associated AEs and resistance. Since the efficacy of conventional cancer therapies are often limited by a dose-dependent toxicity [50,72,82], PCI may represent a rational and promising approach for chemotherapeutic targeting and killing of drug-therapy or multi-therapy-resistant cancer cells. PCI of bleomycin was safe in patients with squamous cell carcinoma or other advanced or recurrent malignancies of the head and neck, torso, and upper limbs [52,83]. A pivotal Phase-II study is currently recruiting patients with inoperable bile duct cancer to assess effectiveness of PCI of gemcitabine complemented by systemic gemcitabine/cisplatin chemotherapy compared to gemcitabine/cisplatin alone [53].

### 3.2. PCI in Immunotherapy

The immune system can play an important role in fighting cancer cells. Immunotherapy, such as checkpoint-inhibition, cytokine therapy, adoptive cell transfer therapy, and therapeutic vaccines all have the potential to induce immune responses that can surveil tumor, suppress growth of or kill cancerous cells, and give the patient a long-lasting immunity that may prevent remissions. However, cancer cells can interfere with the immune system in many ways. In this case, the potential immune-suppressive tumor micro environment (TME) may represent a significant challenge for effective anti-tumor therapies. The TME can be seen as an environment generated by various interactions between cancer cells and immune cells. Cancer cells, as they develop and grow, exploit immune-regulatory mechanisms by interacting with immune cells such as regulatory T and B cells, DCs, and myeloid-derived suppressor cells. Tumors can downregulate protein p53 or other tumor suppressors, downregulate MHC class I or co-stimulatory molecules on APCs, attract immunosuppressive leucocytes, activate CTLA4, PD1, or other co-inhibitory receptors on T cells [4,59,84,85]. The tumor-cell mediated activation of co-inhibitory receptors on T cells directly interferes with T-cell mediated tumor destruction [86], whereas the lack of co-stimulation can lead to T-cell anergy [87]. Additionally, immune escape due to self-tolerance of tumor antigens makes it difficult to target the immune system, notably T cells [88].

While adjuvant immunotherapy with checkpoint inhibitors have found wide application during the last decade, therapeutic cancer vaccination has proven more laborious and less effective. Cancer vaccination aims to stimulate tumor-specific immune responses against delivered antigens. In order to achieve this, one has to overcome hurdles, such as the correct selection of antigens among the plethora of heterogeneously expressed and genetically unstable tumor antigens [4,89] and the use of appropriate adjuvants. To avoid mechanisms of central tolerance in the thymus, it is important to choose immunogenic antigens for vaccination [59]. In this case, neoantigens and viral antigens are not subject to self-tolerance mechanisms and could be used for stronger anti-tumor T-cell responses than regular tumor antigens. Since tumor neoantigens are usually patient-specific, they typically require personalized vaccines, whereas tumor-specific viral antigens could be used for off-the-shelf vaccines.

Cytotoxic T cells and natural killer (NK) cells have shown to make important immunological contributions in fighting tumors. In order for therapeutic vaccines to trigger the generation of tumor-specific CTLs, the MHC class I pathway of antigen presentation needs to be accessed. However, vaccine antigens end up in phagolysosomes of APCs and are presented in the context of MHC class II, which leads to stimulation of CD4 rather than CD8 T cells. One possible approach for CTL activation is to shuffle the antigen across the plasma membrane and, thereby, avoid endocytosis altogether. Another way has been obtained by triggering endosomal escape of the internalized antigen subsequent to the endocytosis or phagocytosis of antigens into APCs. The combination of antigens with a photosensitizer and light can facilitate cytosolic release of the endocytosed antigen by disruption of the endosomal membrane. The now cytosolic antigen can be processed by proteasomes and presented via MHC class I pathway for stimulation of CD8 T-cell responses, which, therefore, overcame the problem of the CD8 deficit after vaccination (Figure 2).

Successful stimulation of tumor-specific immunity by PCI has been demonstrated in several mouse models of cancer. PCI mediated induction of antigen-specific CD8 T-cell proliferation and IFN-γ production [20,22,33,34,35,36], prevention of tumor grafting [34,35], suppression of tumor growth, and improved progression-free survival in mice [35,37]. Studies have demonstrated the mechanism of antigen and photosensitizer uptake in APCs and that, upon application of light, the antigen is released from endosomes or even phagolysosomes into the cytosol [34,35,36]. While it has been recognized that the generation of primary CD8 T-cell responses to non-inflammatory antigens typically require MHC class II-restricted CD4 T helper cells, Varypataki et al. demonstrated that CD8 T-cell responses and their ability to control tumor growth after PCI-based vaccination were not impaired in MHC class-II and CD4 T-cell deficient mice [37]. In order to verify the significance of the data, further tumor models will be needed, and future findings may have clinical importance with regard to the fact that many tumor patients are treated with CD4 T-cell-sensitive immunosuppressive agents [90]. Antigen-specific CD8 T-cell responses could also be generated autologously in mice after prior PCI-mediated loading of the antigen to DCs in vitro [33]. In light of the autologous vaccine Sipuleucel-T, which is the only FDA-approved and therapeutic cancer vaccine, it would be interesting to follow up on this technique with other models, e.g., DNA or mRNA treated DCs or DCs treated with tumor antigens. This would enable us to conclude on the true potential of PCI-based autologous vaccination. In the above mentioned report [33], the DCs were treated with the model antigen OVA, which is a strong antigen, while tumor antigens are typically weak.

A Phase-I clinical trial was completed on 27 August, 2019 on the safety of photochemical internalization of a large immunogenic protein (KLH) and two smaller and less immunogenic peptides (HPV) in healthy volunteers [91]. The primary objective was to study the incidence of AEs after a single administration of the photosensitizer and light. The first results thereof were presented at the ESMO Immuno-Oncology Congress in December 2019 [92]. The induction of HPV-specific immune response in blood showed an increase in the number of healthy donors with HPV-specific CD4+ and CD8+ T-cell responses to PCI-based vaccination compared to baseline levels. Further details and results of the study are expected to be released. However, additional Phase-II and III trials will be needed to investigate the translational potential of current pre-clinical and anecdotal clinical reports.

Therapeutic benefits of anti-cancer vaccines in development are inconclusive. Even with optimized antigen selection and delivery, tumor-intrinsic evasive actions, as well as the lack of understanding of the tumor-microenvironment, pose unforeseeable obstacles. A deeper understanding of the interactions between the immune system and cancer cells will be inevitable for treatment optimization. One possible approach to overcome the immunosuppressant TME has been suggested to be the combination of cancer vaccines and other immunotherapies such as checkpoint inhibition, cytotoxic agents, or classical chemotherapies [59]. Therapeutic vaccination could help prime the immune system to recognize tumor antigens or individualized neoantigens, and the effect of established cancer therapies could, therefore, be improved [36]. Another obstacle for future clinical trials is expected to be the possible shift from self-tolerance to autoimmunity, triggered by immunotherapy combined with local inflammation following treatment. So far, no such effects have been observed with PCI, but, since autoimmunity typically develops slowly, a further assessment will be required. However, the risk of autoimmune reactions is not limited to cancer vaccination, but to all immune-stimulating procedures. Photochemical internalization, as a technology with the potential to enhance therapeutic cancer vaccines, can be seen as a promising tool to optimize anti-cancer immunotherapy.

### 3.3. Strength and Limitations

The strength of this study include the documented comprehensive literature search and adherence to pre-defined inclusion criteria. The main limitations include the possibility of not covering the total body of evidence on this subject and the challenges of interpreting a heterogeneously formed body of evidence. Furthermore, the individualized selection process and the authors’ involvement in the research could represent a possible bias risk.

## 4. Materials and Methods

### 4.1. Eligibility Criteria

We searched for pre-clinical and clinical trial studies that used the concept of PDT either for the internalization of cytotoxic therapeutics into tumor cells or for the internalization of antigens into the cytosol of APCs for anti-cancer vaccination such as PCI immunotherapy. We included papers published from January 2009 to October 2019. No criteria in terms of participants or study design were imposed, and reviews were not included. The primary outcomes of interest were PCI of cytotoxic therapeutics drugs or antigens, suppression of tumor growth for PCI-based delivery of cytotoxic therapeutics, or the activation of CTLs for PCI-based vaccination.

### 4.2. Information Sources and Search

The three databases searched were PUBMED, www.clinicaltrial.org, and The Cochrane Library. The following search terms were used: (1) “photochemical internalization” OR “photochemical internalisation”, (2) “Amphinex” OR “TPCS2a”, (3) “cytosolic delivery” AND “chemotherapy”, (4) “cytosolic delivery” AND “antigen”, (5) “cytosolic targeting” AND “chemotherapy”, (6) “cytosolic targeting” AND “antigen”, (7) “cytosolic delivery” AND “photodynamic therapy”, (8) “cytosolic targeting” AND “photodynamic therapy”, (9) “cytosolic delivery” AND “vaccine”, (10) “cytosolic targeting” AND “vaccine”, (11) “photodynamic therapy” AND “vaccine”, (12) “photodynamic therapy” AND “antigen”, (13) “cytosolic delivery” AND “immunotherapy”, and (14) “cytosolic targeting” AND “immunotherapy”. PUBMED was searched with all 14 terms. The Cochrane library was searched in search fields “title, abstract, keyword” with all 14 search terms. The trial registry www.clinicaltrials.gov was searched for the terms “photochemical internalization” OR “photochemical internalisation” in the search field. Additionally, “other terms” were used with no entry in the search field “countries” and “conditions and diseases”.

Based on the inclusion criteria, 18 papers were included in this systematic review. Additionally, the cited references in the 18 selected papers were screened for further and relevant background information. The last literature search was made on 2 October 2019.

### 4.3. Study Selection

Eligibility assessment was performed in an unblended manner by two reviewers. Disagreements between reviewers were resolved by consensus. The Preferred Reporting Items for Systematic Reviews and Meta-Analysis Protocols (PRISMA-P) were used as a guide for this systematic review. The search was not restricted by study design or by language. No review protocol has been created and the review has not been registered.

## 5. Conclusions

Research on and clinical development of PCI is still at its very beginning. There have been promising reports and the first in-human trials for the use of PCI in the delivery of chemotherapeutics to targeted cancer cells, but further Phase-II and Phase-III trials will be needed. As for the use of PCI in the field of therapeutic vaccination, pre-clinical data has shown the feasibility of the technology in mouse and cell-line models and has paved the way for the first test in humans. In conclusion, PCI can be seen as a novel technology for intracellular delivery of molecules with the possibility to enhance therapeutic effects of some cancer therapies and help prime the immune system for tumor-antigens.

## Figures and Tables

**Figure 1 cancers-12-00165-f001:**
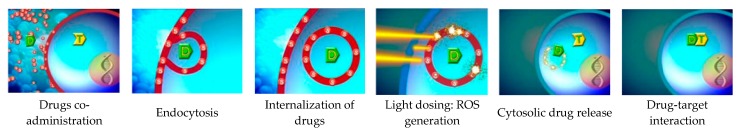
Photochemical internalization. The drug is co-administered with the photosensitizer. The photosensitizer accumulates in cell membranes and the drug is taken up through endocytosis. ROS are generated during illumination, which leads to disruption of the endocytic membrane and release of the drug into the cytosol (modified with courtesy from PCI Biotech: http://pcibiotech.no/what-is-pci/).

**Figure 2 cancers-12-00165-f002:**
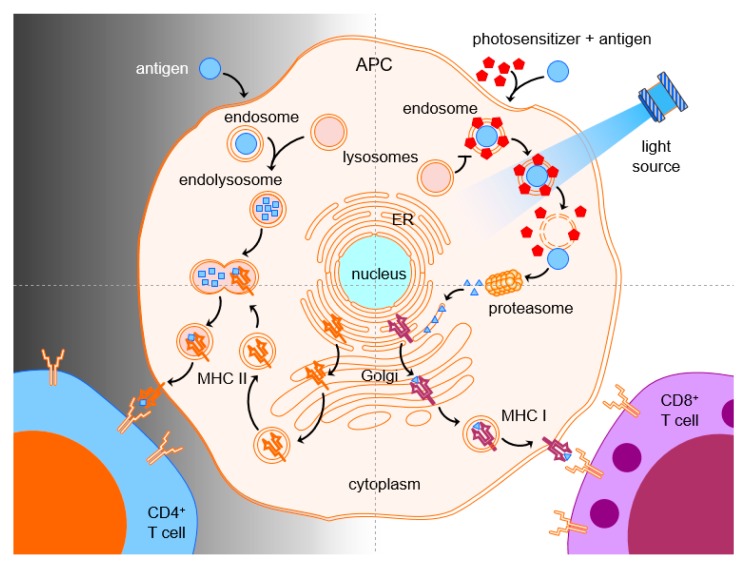
Antigen uptake, processing, and T-cell presentation in PCI-based vaccination. Photosensitizer and antigen are endocytosed into an antigen-presenting cell (APC). The photosensitizer is attached to the endosomal membrane and the antigen is contained in the endosomal lumen. After a wash-out period, where excess photosensitizer dissociates from the outer plasma membrane, light exposure causes endosomal eruption and cytosolic release of antigen for proteasomal degradation and MHC class-I presentation to CD8 T cells. In the absence of the photosensitizer and light, endosomes mature and fuse with lysosomes for MHC class-II presentation of digested antigens to CD4 T cells.

**Figure 3 cancers-12-00165-f003:**
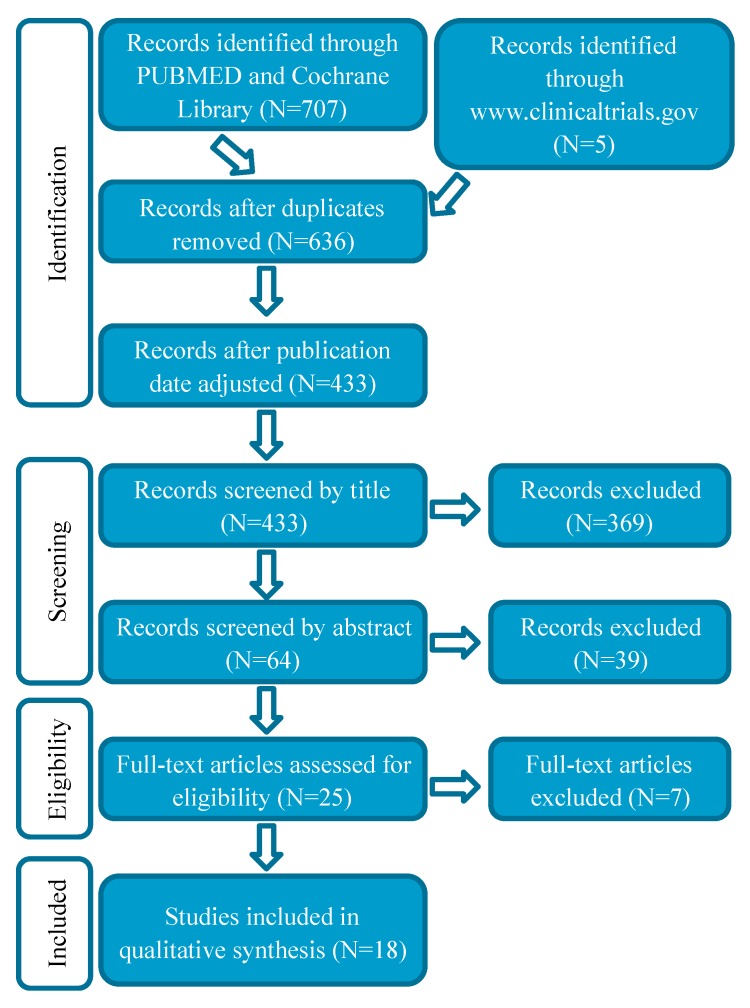
PRISMA flow diagram for systematic selection and review of studies.

**Table 1 cancers-12-00165-t001:** Photosensitizers approved or under clinical trials.

Name	Ex Wave-Length (nm)	Manufacturer	Application
**FIRST GENERATION PHOTOSENSITIZERS**			
Porfimer sodium	630	Axcan Pharma	PDT of esophageal cancer, lung adenocarcinoma, and endobronchial cancer
**SECOND GENERATION PHOTOSENSITIZERS/Prodrugs**			
5-aminolaevulinic acid	635	DUSA Stabiopharma	PDT of mild to moderate actinic keratosis Fluorescence guided resection of glioma
Methyl-aminolevulinic acid	579–670	Galderma	PDT of non-hyperkeratotic actinic keratosis and basal cell carcinoma
Temoporfin	652	Biolitec	PDT of advanced head and neck cancer
Talaporfin	664	Meiji Seika Novartis	PDT of early centrally located lung cancer
Verteporfin	690	Novartis	PDT of age-related macular degeneration
Redaporfin	749	Luzitin	PDT of biliary tract cancer
**PHOTOSENSITIZERS UNDER CLINICAL INVESTIGATIONS**			
Fotolon	665	Apocare Pharma	PDT of nasopharyngeal, sarcoma
Hexylaminolevulinate	635	Photocure	PDT of HPV-induced cervical precancerous lesions and non-muscle invasive bladder cancer
Radachlorin	662	Rada-pharma	PDT of skin cancer
Photochlor (HTTP)	664	Rosewell Park	PDT of head and neck cancer
Padeliporfin	762	Negma-Lerads	PDT of prostate cancer
Motexafin lutetium	732	Pharmacyclics	PDT of coronary artery disease
Rostaprofin	664	Miravant	PDT of age-related macular degeneration
Talaporfin	664	Meiji Seika	PDT of colorectal neoplasms, liver metastasis
Fimaporfin	435	PCI Biotech	PCI of cutaneous or sub-cutaneous malignancies, cholangiocarcinoma and PCI of vaccine antigens

**Table 2 cancers-12-00165-t002:** Articles on PCI of cytotoxic therapeutics reviewed in this paper.

Author, Year, Country	Tested Tissue/Cells	PCI-Internalized Molecule and Photosensitizer	Study Model	Primary Outcome	Reference
**PRECLINICAL STUDIES**					
Olsen et al., 2013	Dox-resistant human sarcoma cell line MES-SA/Dx5 and non-resistant MES-SA line	rGel and TPCS_2a_	In vitro: Cell culture	PCI circumvents the mechanisms of PDT resistance in dox-resistant human sarcoma cell lines	[25]
O’Rourke et al., 2017	Rat cortical mixed glial cells, DRG, and satellite glia, HNSCC cell line	Bleomycin and TPPS_2a_ or TPCS_2a_	In vitro: Cell culture	DRG neurons can survive TPCS_2a_ and TPPS_2a_-mediated PCI at doses enough to kill the carcinoma cell line	[26]
Martínez-Jothar et al., 2019	Human HER2+ and HER2− breast cancer cell lines	Saporin or placebo in PEGylated NP and TPPS_2a_, functionalized with 11A4 nanobody	In vitro: Cell culture	PCI of saporin-loaded PEGylated NP can be used to selectively induce cell death of HER2+ breast cancer cells	[27]
Norum et al., 2017	Murine CC and murine MGC cells in athymic and thymic BALB/c mice	Bleomycin and AlPcS_2a_	In vivo: Murine allograft model	PCI of bleomycin had a curative effect on tumor cells in thymic, but not in athymic mice and induced immune responses sufficient to reject new tumor cells for up to two months	[28]
Stratford et al., 2013	Human sarcoma cell line and human fibrosarcoma cell line, human sarcoma cells in athymic nude mice	CD133-targeting immunotoxins and TPCS_2a_	In vitro: Cell culture In vivo: Murine xenograft model	Proof-of-concept: PCI of CD133-targeting immunotoxins reduces cellular viability and proliferative capacity of sarcoma cells and inhibits tumor grafting	[29]
Bostad et al., 2015	Human CA, ALL, malignant melanoma, and TNBC (CD133+ and CD133−) cell lines, human CA cells in athymic nude mice	CD133-targeting immunotoxin and TPCS_2a_	In vitro: Cell culture In vivo: Murine xenograft model	Efficient PCI of CD133-targeting immunotoxins in human cancer cell lines in vitro. Proof-of-concept: Anti-tumor response after PCI of CD133-targeting immunotoxins in vivo	[30]
Eng et al., 2018	Human TNBC, amelanotic human melanoma, human Melmet cell lines, amelanotic human melanoma cells in athymic nude mice	CSPG4-targeting toxin and TPCS_2a_	In vitro: Cell culture In vivo: Murine xenograft model	PCI of CSPG4-based immunotoxins induces death of CSPG4-positive and drug-resistant cells of TNBC and malignant melanoma origin, in vitro and in vivo	[32]
Berstad et al., 2015	Head and neck squamous cell carcinoma cell line. A-431or SCC-026 cells in athymic nude mice	rGel/EGF (an EGFR-targeted fusion protein)	In vitro: Cell culture In vivo: Murine xenograft model	PCI increased the cytotoxicity of rGel/EGF in EGFR-expressing cells. PCI of rGel/EGF induced significant antitumor effects in A-431 xenograft mice	[31]
Weyergang et al., 2018	VEGFR2-expressing endothelial cells, murine colon carcinoma, and breast carcinoma cell lines in vitro and in BALB/c and athymic nude mice	VEGF_121_/rGel and TPPS_2a_ (in vitro) or TPCS_2a_ (in vivo)	In vitro: Cell culture In vivo: Murine allograft model	PCI of VEGF_121_/rGel directly targets tumor cells and induces T-cell mediated tumor remission, reduced perfusion, and produced tumor protection in vivo	[63]
**CLINICAL TRIALS IN HUMAN**					
Sultan et al., 2016, England	Patients (18 to 85 years) with local recurrent, advanced, or metastatic cutaneous or subcutaneous malignancies	Bleomycin and TPCS_2a_	Phase I, First-in-human	Administration of TPCS_2a_ was found to be safe and tolerable by all patients. No significant systemic adverse events related to photochemical internalization treatment occurred.	[52,64]
Jerjes et al., 2019, England	57-year-old male with end-stage recurrent and therapy-resistant chondroblastic osteosarcoma in the right mandible	Bleomycin and TPCS_2a_	Case report	Illuminated areas responded favorably to treatment. PCI anti-tumor activity was superior to PDT, clinically and histopathologically. Peri-illumination pain	[65]

Dox: Doxorubicine. DRG: dorsal root ganglion neurons. HNSCC: head and neck squamous cell carcinoma. NP: Nanoparticles. CC: Colon carcinoma. MGC: Mammary gland carcinoma. CA: Colorectal adenocarcinoma. ALL: Acute lymphocytic leukemia. TNBC: Triple negative breast cancer. CSPG4: cell surface chondroitin sulfate proteoglycan 4. rGel: recombinant Gelonin. EGFR: epidermal growth factor receptor.

**Table 3 cancers-12-00165-t003:** Articles on PCI in immunotherapy reviewed in this paper.

Author, Year, Country	PCI-Internalized Molecule and Photosensitizer	Study Model	Primary Outcome	Reference
**PCI IN IMMUNOTHERAPY, preclinical studies**				
Waeckerle-Men et al., 2013	OVA and TPCS_2a_	In vitro: DCs In vivo: Autologously immunized C57BL/6 mice	Proof-of-concept: Feasibility of PCI of OVA in DCs for stimulation of CTL responses in vitro. Autologous vaccination of mice with PCI-treated DCs led to improved MHC class-I-restricted and antigen-specific CTL response	[33]
Håkerud et al., 2014	OVA and TPCS_2a_	In vitro: DCs In vivo: Allograft model, B16-OVA-melanoma cells in C57BL/6 mice	Proof-of-concept: Photosensitization and immunization directly in vivo. PCI-vaccination stimulated antigen-specific CD8 memory cells and prevented tumor growth	[34]
Håkerud et al., 2015	OVA and TPCS_2a_	In vitro: DCs In vivo: Allograft model, B16-OVA-melanoma cells in C57BL/6 mice	Prophylactic PCI-based vaccination prevented tumor grafting and therapeutic vaccination reduced tumor growth and improved mouse survival	[35]
Hjálmsdóttir et al., 2016	DPPC Liposomes loaded with OVA or TPCS_2a_	In vitro: TPCS_2a_-and OVA-loaded liposomes Ex vivo CTL in blood and spleen of C57BL/6 mice	Liposomes can be used for PCI-based cytosolic antigen targeting and CTL cross priming and may protect photosensitizers from light-induced inactivation	[20]
Bruno et al., 2015	PLGA microparticles loaded with OVA and TPCS_2a_	In vivo: Allograft model, B16-OVA-melanoma cells in C57BL/6 mice	The combination of PLGA microparticle–based antigen delivery and photosensitization induces stimulation of antigen-specific CTL in a mouse model	[22]
Haug et al.	In vitro: OVA and TPCS_2a._ In vivo: HPV 16 E7 protein (HPV43–78) and from tyrosinase-related protein 2 (TRP2 180–188) and TPCS_2a_	In vitro: Macrophages In vivo: C57BL/6 mice	PCI promotes delivery of peptide antigens to the cytosol of APCs in vitro. Successful induction of antigen-specific CTL responses following intradermal peptide vaccination using PCI in vivo	[36]
Varypataki et al., 2019	OVA and TPCS_2a_ or lethally irradiated B16-OVA and TPCS_2a_	In vivo: Allograft model, B16-OVA-melanoma cells in C57BL/6, congenic CD45.1, MHC class II- and CD40L-deficient mice	PCI-based vaccination caused tumor regression independent of MHC class II or CD4 T cells in melanoma-bearing mice	[37]

OVA: Chicken ovalbumin. DCs: Dendritic cells. DPPC: Dipalmitoyl phosphatidylcholine. PLGA: poly(lactide-co-glycolide).

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
