# Peer review of "Photochemical Internalization: Light Paves Way for New Cancer Chemotherapies and Vaccines"

_cancers, 2020, doi:10.3390/cancers12010165_

Round 1

Reviewer 1 Report

The review by Sosic et al. titled “Photochemical Internalization: Light Paves Way for New Cancer Chemotherapies and Vaccines” is an interesting effort summarizing the outcome of eighteen investigations between January 2009 and October 2019 that described photochemical internalization (PCI) of chemotherapeutic agents or antigens in both pre-clinical and clinical studies, to improve the therapeutic outcome. The authors searched PUBMED, www.clinicaltrial.org, and The Cochrane Library for reviewing the literature published. While the summary of pre-clinical studies suggested that PCI could be successfully used to deliver chemotherapeutic drugs to the tumor cells improving treatment efficacy, more systematic clinical trials are needed to establish the translational potential of PCI in clinic.

This is a well-written review and the overall approach with Introduction, methods (criteria for inclusion and exclusion of studies), results and discussion is satisfactory. The claim made by authors about translational potential of PCI in clinic to eventually improve the therapeutic outcome of PDT seems feasible. However, two minor corrections and changes are suggested to make this manuscript suitable for publication in Cancers.

In the Introduction, the statement “Briefly, a photosensitizer is administered to the tumor lesion. Subsequent light activation, induces photochemical energy, the generation of reactive oxygen species (ROS), and damage the cell membrane to cause cell death” (page 2, line 80) should have a few relevant references from the literature. In the Results, the subsection “2.5. Synthesis of results” (page 6, line 438) doesn’t seem meaningful. Also, the content in the subsection does not relate to the subheading. This subsection should be removed and the content can be placed in other relevant subsections.

Author Response

The authors are grateful for the reviewer's overall positive peer-review of our submission.

As recommended by the reviewer, we have added additional references to the mentioned section in the Introduction discussing the cytotoxic effect of photochemical energy:

Agostinis et al Photodynamic Therapy of Cancer: An Update. Ca Cancer J Clin 2011, 61, 250–281. Juarranz et al. Photodynamic therapy of cancer. Basic principles and applications, Clin Transl Oncol. 2008, 10, 148-154. van Straten et al. Oncologic Photodynamic Therapy: Basic Principles, Current Clinical Status and Future Directions. Cancers. 2017, 9, 18.

Moreover, and as suggested, we have removed the section "2.5. Synthesis of results". Being a intermittent summary, it was indeed redundant to the text preceding it and to final and general conclusion.  As the section did not contain claims or issues that were not discussed elsewhere, the text content of section 2.5 has not been place elsewhere in the submission. 

Reviewer 2 Report

The authors of the manuscript presented a literature review of Photochemical Internalization (PCI) delivery method opening new and promising applications in cancer chemotherapies and vaccines.
Manuscript is a very reliable, detailed review of research and the application of PCI delivery system. It is a very valuable source of information for readers.
I recommend the manuscript for publication in the Cancers journal.

Author Response

The authors are very grateful for the reviewers positive recommendation of our submission.